# Preparation and Immunological Efficacy Evaluation of mRNA Vaccines Targeting the Spike Protein of Bovine Coronavirus

**DOI:** 10.3390/vaccines13111155

**Published:** 2025-11-12

**Authors:** Shuyue Liu, Zhen Gong, Ping Wang, Fu Chen, Xiulong Fu, Haoyu Fan, Yue Li, Xiangshu Han, Junli Chen, Lixue Zhang, Lijun Xue, Hangfei Bai, Shufan Liu, Lulu Huang, Wei Du, Ang Lin, Jun Xia

**Affiliations:** 1College of Veterinary Medicine, Xinjiang Agricultural University, Urumqi 830052, China; 18690235702@163.com (S.L.); fxl0002152022@163.com (X.F.);; 2Veterinary Rasearch Institute Xinjiang Uygur Autonomous Region Academy of Animal Husbandry Sciences (Xinjiang Uygur Autonomous Region Academy of Animal Husbandry Sciences Center for Clinical Veterinary Medicine), Urumqi 830010, China; 15029968328@163.com (Z.G.);; 3Xinjiang Animal Disease Control and Prevention Engineering Technology Research Center, Urumqi 830000, China; 4College of Animal Science and Technology, Shihezi University, Shihezi 832003, China; fanhy148@126.com (H.F.);; 5College of Veterinary Medicine, Nanjing Agricultural University, Nanjing 210095, China; 6Vaccine Center, School of Basic Medicine and Clinical Pharmacy, China Pharmaceutical University, Nanjing 211198, China; 7Livestock Research Institute Xinjiang Uygur Autonomous Region Academy of Animal Husbandry Sciences, Urumqi 830099, China; 8Center for New Drug Safety Evaluation and Research, China Pharmaceutical University, Nanjing 211198, China

**Keywords:** bovine coronavirus, mRNA vaccine, humoral immunity, cellular immunity

## Abstract

Objectives: Bovine coronaviruses (BCoV) are endemic worldwide, causing diarrhea, winter dysentery, and bovine respiratory disease in newborn calves. These lead to higher calf mortality, reduced growth of fattening cows, and lower milk production in adult cows, resulting in significant losses to the cattle industry. Since commercial preventive drugs are not available in China, and existing treatments can only reduce the mortality of sick calves without fundamental control, the development of safe and effective vaccines is crucial. Methods: Two mRNA vaccines targeting the BCoV spiny receptor-binding domain (S-RBD) were prepared: XBS01 and XBS02. These two mRNAs, optimized for coding by AI and encapsulated in lipid nanoparticles (LNPs), were injected intramuscularly into mice (10 μg per mouse, twice, 2 weeks apart); a blank control group was not immunized. Serum antibodies, memory B/T cell activation and cytokine secretion were assessed by ELISA, flow cytometry and ELISpot. Results: Both vaccines induced humoral and cellular immunity:anti-S-RBD IgG titers were higher than those of the control group, and there was memory B-cell production and T-cell activation. XBS02 was superior to XBS01 in terms of peak antibody, memory B-cell frequency, T-cell activation rate, and IFN-γ/IL-2 secretion, and showed a stronger Th 1 response. Conclusions: Both BCoV S-RBD mRNA vaccines had good immunogenicity, with XBS02 providing better protection. This study supports the optimization and application of BCoV mRNA vaccines and accumulates data for mRNA technology in veterinary practice.

## 1. Introduction

Bovine Coronavirus (BCoV) was first identified in cattle farms in the United States in 1973, followed by outbreaks in Europe and Asia [1]. In recent years, BCoV prevalence has been reported in countries across all five continents [2]. Due to its rapid transmission via the fecal-oral and respiratory routes, combined with the presence of asymptomatic infections in cattle herds, BCoV has become widespread globally [3]. BCoV infects the respiratory and intestinal tracts of cattle and wild ruminants and is a key pathogen causing neonatal calf diarrhea, winter dysentery, and bovine respiratory disease [4]. These three syndromes increase calf mortality, reduce the growth performance of fattening cattle, and decrease milk production in adult dairy cows [5,6], leading to significant economic losses. Currently, there are no commercialized specific preventive drugs for BCoV-induced calf diarrhea in China. Clinical management typically involves supportive care, symptomatic treatment, and traditional Chinese medicine therapy to help infected calves replenish fluids and electrolytes, prevent dehydration and acidosis, reduce secondary infections, and thereby lower mortality. Currently, there are studies on the trivalent vaccine against bovine respiratory syncytial virus, bovine parainfluenza virus type 3, and bovine coronavirus, as well as studies on the bovine coronavirus vaccine using LSDV as a live viral vector, but none of these have been widely produced. Therefore, to prevent BCoV-induced calf diarrhea, this study designed and initially prepared BCoV mRNA vaccines and verified their immunological efficacy in mice to provide a reference for BCoV prevention.

BCoV belongs to the genus Betacoronavirus in the family Coronaviridae. It is an enveloped, single-stranded positive-sense RNA virus with a genome size of 22–36 kb. Virions are polygonal or spherical, with a diameter of 65–210 nm, and contain 10 open reading frames (ORFs) [7]. The 5′ end of the genome contains two large ORFs (1a and 1b), which encode polyproteins involved in viral replication [8]. BCoV has five main structural proteins: nucleocapsid protein (N protein), membrane glycoprotein (M protein), envelope protein (E protein), hemagglutinin-esterase protein (HE protein), and spike protein (S protein) [9]. The coronavirus S protein consists of three main regions: an extracellular domain, a transmembrane domain, and an intracellular domain [10]. During coronavirus maturation, the extracellular domain of the S protein is cleaved by host proteases into two subunits, S1 and S2 [11]. BCoV invades host cells by binding the S1 subunit of its S protein to the host cell membrane receptor (N-acetyl-9-O-acetylneuraminic acid) [12], while the S2 subunit mediates the fusion of the viral envelope with the host cell membrane, enabling viral entry into the host cell [13]. Current research divides the coronavirus S1 subunit into two structural domains: the N-terminal binding domain (S1-NTD) and the C-terminal binding domain (S1-CTD), both of which can function as receptor-binding domains (RBDs) [14]. These domains are primarily responsible for recognizing coronavirus receptors and mediating the binding of virions to receptors [15]. To date, most coronavirus research has focused on RBDs, as they contain major neutralizing epitopes, exhibit strong immunogenicity [16,17], and can induce host immune responses. Thus, RBDs are the preferred targets for developing antibodies, vaccines, and antiviral drugs.

mRNA vaccine technology is an innovative approach to vaccine development that combines molecular biology and immunology, offering multiple advantages over traditional vaccines. The mechanism of mRNA vaccines involves introducing exogenous mRNA encoding antigenic proteins into somatic cells [18]; transfected non-immune cells then produce the desired antigens. These antigens are subsequently degraded in cytoplasmic proteasomes, and antigenic epitopes complexed with major histocompatibility complex (MHC) class I molecules are presented to antigen-presenting cells [19], facilitating the establishment of cellular immune responses against the mRNA-encoded antigen [20]. Owing to its unique technical advantages, mRNA vaccine technology has achieved remarkable success in disease prevention and control, providing a new strategy for BCoV prevention. During the COVID-19 pandemic, mRNA vaccines (e.g., those developed by Pfizer/BioNTech and Moderna) demonstrated exceptional speed in development and protective efficacy: clinical trials were initiated within months of the release of the viral genome sequence, and large-scale application effectively reduced the rates of severe illness and mortality, verifying the rapid response capability and immunogenicity of mRNA technology [21]. For BCoV, mRNA vaccines also offer significant advantages: they can be rapidly updated to match new viral strains by adjusting sequences based on genetic mutations, addressing the lag of traditional vaccines in responding to strain variations [22]; meanwhile, intracellular antigen synthesis enables efficient activation of both humoral and cellular immunity, inducing more durable and comprehensive immune protection [23]. Therefore, research on BCoV mRNA vaccines is expected to break through existing bottlenecks in prevention and control technology, advance the precision and efficiency of vaccine development, and play a critical role in ensuring the healthy and stable development of the cattle industry. This research will significantly reduce economic losses caused by BCoV and support the sustainable development of animal husbandry.

## 2. Materials and Methods

### 2.1. Main Reagents 

DNA purification kits were purchased from Takara Bio Inc. (Kusatsu, Japan); T7 RNA polymerase; Qubit RNA HS Assay Kit and Qubit™ Assay Tubes and RNA quantification kits from Thermo Fisher Scientific Inc. (Waltham, MA, USA); RNA purification kits from New England Biolabs Inc. (Ipswich, MA, USA); SM-102, DSPC, cholesterol, and mPEG-DMG from SINOPEG BIOTECH Co., Ltd. (Xiamen, China); 10× IVT Reaction Buffer was purchased from SYNTHGENE BIOTECHNOLOGY Co., Ltd. (Nanjing, China); ribonucleotide substrates, Inorganic Pyrophosphatase, RNase Inhibitor, and Cap 1 cap analog were purchased from Hongene Biotechnology Co., Ltd. (Shanghai, China); 10× TBE buffer and bovine serum albumin (BSA) from Solarbio Science & Technology Co., Ltd. (Beijing, China); HRP-conjugated goat anti-mouse IgG from Beyotime Biotechnology Inc. (Shanghai, China); and flow cytometry antibodies from BIOLEGEND (Beijing) Biotechnology Co., Ltd. (Beijing, China).

### 2.2. Experimental Animals

Male BALB/c mice (5 weeks old) were purchased from the Animal Experiment Center of Xinjiang Medical University (production license number: SCXK 2023-0001). The mice were randomly divided into 3 groups (12 mice per group) designated as XBS01, XBS02, and blank control.

### 2.3. mRNA Vaccine Preparation

Two mRNA constructs targeting the BCoV S protein receptor-binding domain (S-RBD) were designed: XBS01 and XBS02 (The specific sequences are provided in Appendix A). Codon optimization was performed using an artificial intelligence (AI)-based algorithm platform (LinearDesign) to construct and synthesize plasmids. The linearized DNA encoding the bovine coronavirus (BCoV) S-RBD immunogen was used for in vitro transcription to synthesize mRNA by mixing 10× IVT Reaction Buffer, ribonucleotide substrates, Cap1 analog, T7 RNA polymerase, inorganic pyrophosphatase, RNase Inhibitor, and RNase-free Water, followed by incubation at 37 °C for 2 h. After purification using an RNA purification kit (NEB), resuspend the purified RNA in TE buffer to a final volume of 50 μL. For encapsulating mRNA into lipid nanoparticles (LNPs), lipid components (SM-102: DSPC: cholesterol: mPEG-DMG) were dissolved in ethanol at a molar ratio of 50:10:38.5:1.5. The lipid mixture was combined with mRNA dissolved in 10 mM citrate-sodium citrate buffer (pH 4.0) using a microfluidic device, The ratio of the aqueous phase to the alcoholic phase is 3:1, and the total flow rate is 12 mL/min. After a 10-fold dilution with PBS, the formulation was ultrafiltered using a 50 kDa ultrafiltration centrifugal filter to remove unincorporated free lipids and residual organic solvents from the lipid mixture. Using the Qubit RNA HS Assay Kit, mix the LNP sample with Triton X-100 and TE buffer (provided in the kit) at a ratio of 2:1:1, followed by incubation at 37 °C for 10 min. The resulting solution was used to determine the concentration of encapsulated mRNA (serving as the total mRNA concentration for calculation). For the free mRNA concentration in LNP, the LNP sample was used without any treatment. Prepare the working solution at a 1:200 ratio. Add the pre-diluted RNA standards (concentration range: 100 ng/μL) to Qubit tubes, respectively, then add the samples (at the desired concentrations) and the corresponding volume of working solution to each Qubit tube to a final volume of 200 μL. After thorough mixing, place the tubes into the Qubit fluorometer for encapsulation efficiency detection [Encapsulation efficiency (%) = (Total RNA amount − Free RNA amount)/Total RNA amount × 100%]. The liposomes were diluted to a concentration of 0.1 mg/mL with PBS, allowed to stand for 5 min, and after removing bubbles, the sample was slowly injected into a capillary sample cell dedicated to ZETA potential measurement and placed on the instrument sample stage. In the instrument software, the parameters were set as follows: medium properties: water, viscosity 0.8936 mPa·s, refractive index 1.33; temperature: 25 °C; electric field parameters: 40–50 V, 12 sub-tests. After the detection was completed, the instrument automatically calculated the ZETA potential value and distribution curve.

### 2.4. Immunization Protocol

Mice in each group were immunized twice via intramuscular injection (i.m.) (Figure 1, with a dose of 200 μL per mouse (containing 10 μg of mRNA) and a 2-week interval between immunizations. Serum samples were collected 7 days after each immunization for ELISA analysis. At specified time points post-vaccination, 6 randomly selected mice from each group were euthanized; spleens and draining lymph nodes (dLNs) were collected and processed into single-cell suspensions for ELISpot, memory B cell (MBC), and activation-induced marker (AIM) T cell analysis. The remaining 6 mice were subjected to weekly blood collection to monitor changes in antibody levels.

### 2.5. Indicators and Methods for Immunological Efficacy Detection

#### 2.5.1. ELISA Antibody Detection

RBD protein was coated onto 96-well plates at a concentration of 100 ng/well and incubated overnight at 4 °C. Plates were washed three times with PBS containing 0.075% Tween-20 (PBST) and blocked with 2% BSA at 37 °C for 2 h. Positive serum from vaccinated mice was used as the primary antibody: serum collected after the first immunization was serially diluted from 1:50 to 1:51, 200, and serum collected after the second immunization from 1:200 to 1:409, 600, followed by incubation at 37 °C for 2 h. For mouse sample analysis, HRP-conjugated goat anti-mouse IgG (1:50,000 dilution) was used as the secondary antibody and incubated at 37 °C for 1 h to detect bound IgG. TMB substrate was added for color development for 15 min; the reaction was stopped, and absorbance was measured at 450 nm. The endpoint titer was calculated as the dilution factor at which the measured optical density (OD) value was 4.1 times higher than the background value. IgG1 and IgG2c were detected using the same method.

#### 2.5.2. Memory B Cell Response Analysis

Flow cytometry was used to evaluate the frequency of BCoV RBD-specific class-switched IgD^−^IgM^−^ memory B cells (MBCs). For RBD probe preparation, biotinylated RBD protein was conjugated with BV421- and APC-labeled streptavidin at a molar ratio of 4:1. Cells isolated from mouse spleens and draining lymph nodes (dLNs) were first incubated with the RBD probe for 20 min, then stained with fixable eFluor™ 506 (Thermo Fisher Scientific) for 5 min. After washing, cells were incubated with Fc receptor blocking reagent (Miltenyi, Shanghai, China) and an antibody mixture at 4 °C in the dark for 20 min. Flow cytometry analysis was performed on a BD FACSymphony A3 (BD Biosciences, Franklin Lakes, NJ, USA), and data were analyzed using FlowJo V.10.1 software.

#### 2.5.3. Antigen-Specific T Cell Detection

To evaluate RBD-specific T cell responses, 3 × 10^6^ mouse splenocytes per well were seeded into 96-well U-bottom plates. One group was incubated with RBD protein (5 μg/mL) for 24 h, another group without protein, and the positive control group with Staphylococcal Enterotoxin B (SEB: 1 μg/mL). Cells were then stained with fixable eFluor™ 506 (Thermo Fisher Scientific) for 5 min, washed, and incubated with an antibody mixture at 4 °C in the dark for 20 min. Flow cytometry analysis was performed on a BD FACSymphony A3, and data were analyzed using FlowJo V.10.1 software.

#### 2.5.4. Enzyme-Linked Immunospot (ELISpot) Detection

Coating antibody was diluted to 15 μg/mL with PBS, and 100 μL was added per well for overnight incubation at 4 °C. Mouse splenocytes were prepared and adjusted to a concentration of 2 × 10^6^ cells/mL. Splenocytes (2 × 10^5^ cells per well) were co-incubated with RBD protein (5 μg/mL) for 24 h; the negative control group received 100 μL of 1640 medium, and the positive control group 100 μL of SEB (1 μg per well), with 2 replicate wells per group. After incubation, the medium was discarded, the plates were washed 5 times with PBS and patted dry. Primary antibody (1 μg/mL) was added at 100 μL per well and incubated at room temperature for 2 h. After discarding the primary antibody, plates were washed 5 times with PBS and patted dry, followed by incubation with secondary antibody (1 μg/mL) for 1 h. After washing 5 times, 100 μL of filtered BCIP/NBT-plus chromogenic solution was added per well, and color development was monitored in the dark. After color development, plates were rinsed thoroughly under running water, air-dried, and stored at room temperature in the dark. Plates were read using an ELISpot reader, and spots were counted using a CTL-Immunospot S6 analyzer. Results were expressed as the number of spot-forming cells (SFCs) per million stimulated cells.

## 3. Results

### 3.1. Design and Characterization of BCoV mRNA Vaccines (XBS01 and XBS02)

The Spike Protein Receptor-Binding Domain (S-RBD) is the major immunogen of BCoV and has been widely used as a key antigen candidate for BCoV vaccine development. The mRNA vaccines designed in this study encode the RBD antigen, as RBD contains major neutralizing epitopes, exhibits strong immunogenicity, and can induce host immune responses. Based on the S-RBD, mRNA vaccines were designed and prepared, named XBS01 and XBS02. A proprietary AI-based algorithm (LinearDesign) was used to optimize mRNA sequences for optimal folding stability and codon usage, thereby improving translation efficiency and vaccine immunogenicity. Using this AI tool, codon-optimized RBD-mRNA sequences were designed based on S-RBD. The optimized RBD-mRNA was synthesized via in vitro transcription (IVT) with N1-methylpseudouridine (m1Ψ) modification. BCoV-mRNA vaccine formulations (designated XBS01 and XBS02) were further prepared by encapsulating RBD-mRNA into a lipid nanoparticle (LNP) delivery system using a microfluidic technology-based method. Dynamic light scattering analysis confirmed high uniformity in the shape and size of the nanoparticles. Dynamic light scattering showed particle sizes of 79.68 nm (XBS01) and 78.24 nm (XBS02). The ZETA potentials were as follows: 1.77 mV (XBS01) and 3.95 mV (XBS02), with an encapsulation efficiency of 89% for both. The particle size distribution (PDI) was all less than 0.2 (Figure 2a–d).

### 3.2. Induction of BCoV S-RBD-Specific Antibodies by Vaccines in Mice

ELISA (Figure 3a–c), a classic method for evaluating humoral immunity, clearly showed the dynamic changes in antibody responses post-vaccination. The results indicated that after vaccination with XBS01 or XBS02, the titer of anti-BCoV S-RBD IgG antibodies was significantly higher than that in the PBS control group, and continued to increase from Day 7 post-vaccination, remaining at a high level until Day 42. This result directly confirms that both vaccines effectively activate the differentiation of B cells into plasma cells, thereby initiating and sustaining humoral immune responses. However, inter-group differences gradually emerged during the antibody response: the peak antibody titer induced by XBS02 (occurring on Day 21 post-vaccination) was significantly higher than that of XBS01, demonstrating a stronger capacity to stimulate antibody secretion. Further analysis of antibody subtypes (IgG1/IgG2c) and their dynamic changes, as well as the IgG2c/IgG1 ratio, revealed functional biases in the immune responses induced by the vaccines. The XBS02 group exhibited higher IgG2c levels and a higher IgG2c/IgG1 ratio than the XBS01 group, suggesting that XBS02 is more likely to activate Th1-type CD4^+^ T cells. Activation of Th1-type CD4^+^ T cells provides critical co-signals for the cytotoxic function of subsequent CD8a^+^ T cells. This synergizes with CD8a^+^ T cell activation data (higher frequency of OX40^+^CD137^+^), enabling the vaccine to not only neutralize free viruses but also more efficiently eliminate virus-infected cells, establishing a more comprehensive immune defense system against viral infection.

### 3.3. Induction of Specific S-RBD Memory B Cell Responses by Vaccines in Mice

Memory B cells (MBCs) are core mediators of long-term vaccine protection; their differentiation and tissue distribution directly determine the efficiency and strength of secondary immune responses. Using RBD-APC labeling detection (Figure 4a,b), dynamic changes in memory B cells in different tissues post-vaccination were clearly observed. The results showed that after vaccination with XBS01 or XBS02, the proportion of RBD^+^ MBCs in the spleen and lymph nodes (dLNs) was significantly higher than that in the control group, confirming that both vaccines drive B cell differentiation into memory phenotypes. When the body is re-exposed to the same antigen, these memory B cells can rapidly activate, proliferate, and differentiate into plasma cells, secreting large quantities of antibodies to initiate an efficient secondary immune response. Further analysis of inter-group differences revealed that XBS02 induced a higher frequency of MBCs in both the spleen and lymph nodes—key immune tissues. The more significant enrichment of MBCs in the lymph nodes of the XBS02 group may be associated with its linked IL-6 sequence, which correlates with more efficient antigen presentation and optimized T-B cell co-activation mechanisms. Antigens are more rapidly and effectively taken up, processed, and presented to T and B cells by antigen-presenting cells, facilitating more robust T-B cell interactions and driving greater B cell differentiation into memory B cells. Meanwhile, the high proportion of MBCs in the XBS02 group’s spleen—an organ critical for maintaining systemic immune memory—provides strong support for the persistence of long-term immune memory. Activated CD4^+^ T cells secrete cytokines that create a suitable microenvironment for B cell differentiation, promoting more B cells to differentiate into memory B cells and thereby enhancing the quantity and quality of the body’s immune memory reserve.

### 3.4. Induction of S-RBD-Specific T Cell Responses by Vaccines in Mice

The results showed that after stimulation with the S-RBD antigen, the frequency of OX40^+^CD137^+^CD4^+^ T cells in both the XBS01 and XBS02 groups was significantly higher than that in the unstimulated groups (Figure 5a,b). This indicates that the S-RBD antigen effectively activates CD4^+^ T cells, inducing the expression of co-stimulatory molecules OX40 and CD137 and thereby triggering immune responses. From this perspective, the antigen stimulation exhibits good immunogenicity and can initiate antigen-specific immune responses. Comparing the XBS01 and XBS02 groups, the frequency of OX40^+^CD137^+^CD4^+^ T cells induced by XBS02 after stimulation was significantly higher than that by XBS01, suggesting that XBS02 may be more effective in activating CD4^+^ T cells and promoting immune responses. It may more efficiently mobilize CD4^+^ T cell-mediated immune reactions and exhibit superior performance in assisting the activation of other immune cells and regulating the intensity of immune responses.

In experiments gating on CD3^+^CD8a^+^ cells, the frequency of OX40^+^CD137^+^CD8a^+^ T cells in the PBS group remained low in both stimulated and unstimulated states. In contrast, after S-RBD antigen stimulation, the frequency of these cells increased significantly in the XBS01 and XBS02 groups. This similarly confirms that the S-RBD antigen activates CD8a^+^ T cells, inducing the expression of OX40 and CD137 co-stimulatory molecules and triggering cellular immune responses, indicating that the antigen can induce the body to generate antigen-specific cytotoxic T cells with the ability to initiate immune responses. Comparative analysis showed that the frequency of OX40^+^CD137^+^CD8a^+^ T cells induced by XBS02 after stimulation was significantly higher than that by XBS01, implying that XBS02 is more effective in activating CD8a^+^ T cells. Since CD8a^+^ T cells play a key role in directly killing pathogen-infected cells and tumor cells, XBS02 may have greater potential in eliminating infected cells and inhibiting the replication and spread of pathogens in the body. However, similar to the CD4^+^ T cell experiment, this evaluation only focuses on the activation of specific T cell subsets; a comprehensive understanding of the vaccine’s immune protective effects requires integration with other immune indicators.

### 3.5. Enhanced Cytokine Secretion by Mouse Splenocytes Induced by Vaccines

This experiment focused on evaluating the induction of IFN-γ and IL-2 secretion by immune cells under different treatment conditions(Figure 6a,b). Four sample groups were set up: PBS, XBS01, XBS02, and SEB, with two conditions (S-RBD stimulation and no stimulation [Unstim]). The results showed that under S-RBD stimulation, the number of IFN-γ and IL-2 spots in the XBS02 group was significantly higher than that in the XBS01 and PBS groups, indicating that XBS02 has a stronger ability to induce immune cells to secrete these two cytokines. The SEB group, as a positive control, showed a high number of spots, verifying the validity and sensitivity of the experimental system; the unstimulated groups showed very few spots, confirming that S-RBD stimulation plays a critical role in inducing cytokine secretion. In conclusion, XBS02 induces higher levels of IFN-γ and IL-2, implying that it can more efficiently activate Th1-type cellular immune responses. These two cytokines synergistically enhance the intracellular clearance of BCoV and long-term immune protection, which is one of the key mechanisms underlying XBS02’s superior immunogenicity over XBS01. This may be attributed to XBS02’s more optimized mRNA sequence design, enabling more efficient expression of the BCoV S-RBD protein and thus more effective activation of antigen-presenting cells and T cells. However, since the study only detected two cytokines (IFN-γ and IL-2), and immune responses involve the synergistic action of multiple cytokines and immune cells, future research should expand the detection indicators to comprehensively evaluate the immune response profile induced by the vaccine.

## 4. Discussion

Since its discovery, BCoV has spread widely in many countries worldwide [24,25,26] and has been successively reported in multiple provinces in China [27]. The intestinal and respiratory diseases it causes in cattle herds have become increasingly severe, resulting in enormous economic losses to the cattle industry [28]. Therefore, developing BCoV-related vaccines to prevent this disease is urgent.

The RNA genome of BCoV exhibits a high mutation rate and low fidelity in hosts, enabling the virus to evade the host immune system more easily [29] and posing significant challenges to BCoV prevention and control [30]. Compared with traditional inactivated vaccines, although inactivated vaccines have mature production processes and good safety, they have limitations such as long production cycles, the need for multiple inoculations, induction of only humoral immunity, and difficulty in coping with viral mutations. DNA vaccines possess excellent stability and simple preparation characteristics, but their immunogenicity is relatively weak, and they carry potential risks of genomic integration. Viral vector vaccines exhibit good immunogenicity and support single-dose inoculation, but the optimization process of vector construction is relatively complex, with extremely high requirements for vector modification and quality control; otherwise, they may trigger pre-existing immunity or severe adverse reactions. mRNA vaccines demonstrate stronger immunogenicity and can simultaneously activate humoral and cellular immunity to synergistically eliminate pathogens. In addition, their rapid preparation makes them particularly suitable for highly variable viruses. However, due to high production costs, strict storage and transportation conditions, and room for improvement in LNP delivery—requiring continuous attempts to use different lipid encapsulants to improve delivery efficiency [31]—they are currently not suitable for large-scale production of veterinary vaccines. Notably, mRNA vaccine technology has not yet been applied to ruminants such as cattle, pointing out directions for future development. In summary, commercial vaccines currently on the market are still dominated by inactivated vaccines, and the research and development of novel vaccines face significant challenges. For example, in this study, no additional design controls were conducted for lipid encapsulants, no detailed grouping was performed for vaccine injection doses, and the mRNA design approach was only a new attempt. Although antibody levels were verified in mice, validation in the target animal (cattle) has not been conducted, resulting in a lack of more comprehensive and systematic experimental data in this study. Therefore, subsequent research will be further expanded to return to animal experiments, explore more comprehensive data, and optimize mRNA sequences and lipid encapsulants.

This study focused on the XBS01 and XBS02 vaccines, using multi-dimensional experimental approaches (ELISA antibody detection, memory B cell analysis, and CD4^+^/CD8a^+^ T cell activation phenotype detection) to explore the complex processes of immune responses induced by the vaccines from the perspectives of humoral immune responses, immune memory establishment, and cellular immune synergy, providing a theoretical basis for vaccine development and application. Detailed discussions are as follows:

During the experiment, the body weight of mice was measured once a week. During this period, we found that there was no significant difference in body weight between the vaccine-immunized groups and the blank control group. Additionally, the mice showed no obvious signs of lethargy during the immunization period, nor did they exhibit hair loss at the injection site. Therefore, evaluating the vaccine toxicity solely based on body weight and mental state, the vaccine we designed and prepared may not have toxicity or severe side effects. In terms of antibody titer, T cell activation frequency, and memory B cell proportion, XBS02 outperformed both XBS01 and the PBS control group. Increased antibody titers indicate the body’s ability to produce more antigen-binding antibodies, enhancing viral neutralization; higher T cell activation frequency means more T cells participate in immune responses, strengthening the cytotoxicity of cellular immunity [32]; and a higher proportion of memory B cells provides greater reserves for long-term immune memory. These data fully demonstrate that XBS02’s antigen design is more compatible with the recognition and activation mechanisms of the body’s immune cells, enabling it to induce a stronger immune response than XBS01 and demonstrating significant advantages in the initial stage of immune stimulation. By inducing a Th1-biased antibody subtype (IgG2c dominance) and a high activation state of CD8a^+^ T cells [33], XBS02 successfully establishes dual protective potential through humoral and cellular immunity. During viral infection, free viruses in body fluids can be neutralized by antibodies such as IgG2c, preventing further cell infection; for virus-infected cells, CD8a^+^ T cells exert cytotoxic effects to eliminate them [34]. This balance and synergy in immune responses are highly adapted to the immune defense needs of viral infections, enabling the body to respond to viral invasion more comprehensively and effectively.

The enrichment of memory B cells in the spleen and lymph nodes lays a solid foundation for long-term immune protection. Combined with the potential impact on T cell memory (e.g., central memory T cells [Tcm] and effector memory T cells [Tem], which were not fully verified in this study but can be reasonably inferred from immune response logic), XBS02 is expected to provide more durable protection. When the body is re-exposed to the virus a long time after vaccination, memory B cells and memory T cells can respond rapidly [34], initiating a secondary immune response to quickly clear the virus and effectively prevent disease occurrence and progression, providing long-term and stable immune protection for the body.

As core components of cellular immunity, CD4^+^ and CD8a^+^ T cells’ activation phenotypes (OX40^+^CD137^+^ frequency) further verify the vaccine’s immunogenicity from the perspective of cellular immunity and form a tight synergy with the humoral immune response process. On one hand, activated CD4^+^ T cells secrete cytokine signals necessary for B cell differentiation, supporting the differentiation of B cells from naive to plasma cells [35] and thereby promoting antibody secretion to facilitate humoral immune responses. On the other hand, CD4^+^ T cells can transmit activation signals to CD8^+^ T cells, assisting CD8^+^ T cells in exerting cytotoxic functions [36]. XBS02 induces a higher frequency of OX40^+^CD137^+^CD4^+^ T cells, indicating that this vaccine can stimulate stronger synergy between humoral and cellular immunity assistance. Under its regulation, humoral and cellular immune responses function more efficiently and coordinately. XBS02 also induces a higher frequency of OX40^+^CD137^+^ in CD8a^+^ T cells, which correlates with the dominance of the IgG2c subtype in humoral immunity [37,38,39]. IgG2c antibodies can label infected cells, and activated CD8a^+^ T cells recognize these labels to exert cytotoxic effects, collectively enhancing the intracellular immune defense process from infected cell recognition to elimination. This allows the body to clear intracellular infected cells more accurately and efficiently during viral infection.

In summary, this study employed multi-dimensional experimental approaches, including ELISA, memory B cell analysis, and CD4^+^/CD8a^+^ T cell detection to evaluate the advantages of the two vaccines in eliciting immune responses from different perspectives. For the XBS01 vaccine, the bovine coronavirus (BCoV) S protein signal peptide and Qα sequence were selected for incorporation. Previous studies have shown that Qα addition can increase the production of multiple structural proteins (S, M, and N), accessory proteins (NSP2, NSP16, and ORF3), as well as host cell gene products such as IL-2, IFN-γ, ACE2, and NIBP [40,41,42]. Mechanistically, Qα can enhance mRNA synthesis/stability and promote protein expression and secretion [43,44,45,46]. The BCoV S protein signal peptide can guide the secretory expression of BCoV S protein and its derived RBD antigen in eukaryotic cells. The synergistic effect of these two components significantly increased the antibody titer, memory B cell frequency, and CD4^+^/CD8a^+^ T cell frequency of XBS01 compared with the PBS control group. For the XBS02 vaccine, interleukin (IL)-6 and bovine IgG Fc sequences were incorporated. The signal sequence of IL-6 is widely used in protein expression [47]. A significant positive correlation exists between the affinity of the IL-6 signal peptide for the SRP54M subunit and the translation efficiency of antigenic proteins [48,49,50], thereby enhancing antigen expression levels and inducing stronger humoral and cellular immunity. The bovine IgG Fc sequence can bind to Fc receptors on host cells to prevent rapid clearance of the RBD antigen, prolonging its in vivo persistence. Meanwhile, the Fc fragment can bind to receptors on the surface of antigen-presenting cells to promote antigen phagocytosis, processing, and presentation. The coordinated action of these two components explains why XBS02 slightly outperformed XBS01 in all detection indicators. In BALB/c mice, XBS02 exhibited stronger immunogenicity than XBS01. This research provides valuable theoretical guidance and practical reference for subsequent vaccine development, optimization, and clinical application.

## 5. Conclusions

In conclusion, our experiments verified that both BCoV mRNA vaccines exhibit good immunogenicity in mice and can stimulate immune responses. This further confirms the great potential of mRNA technology in BCoV prevention and control, providing new ideas and directions for the development of veterinary vaccines. To summarize, the XBS01 and XBS02 vaccines each have advantages in BCoV immune prevention and control, and their optimization may provide more comprehensive and long-lasting protection for cattle herds. This study lays a foundation for the development and application of BCoV mRNA vaccines and provides valuable references for the further development of mRNA technology in the field of veterinary vaccines.

## Figures and Tables

**Figure 1 vaccines-13-01155-f001:**
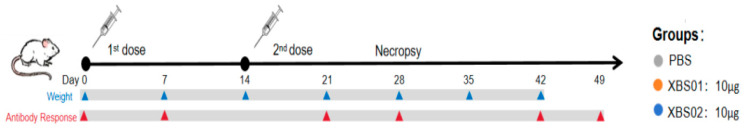
The immunization schedule shows the vaccination protocol for the XBS01 and XBS02 vaccine experiments.

**Figure 2 vaccines-13-01155-f002:**
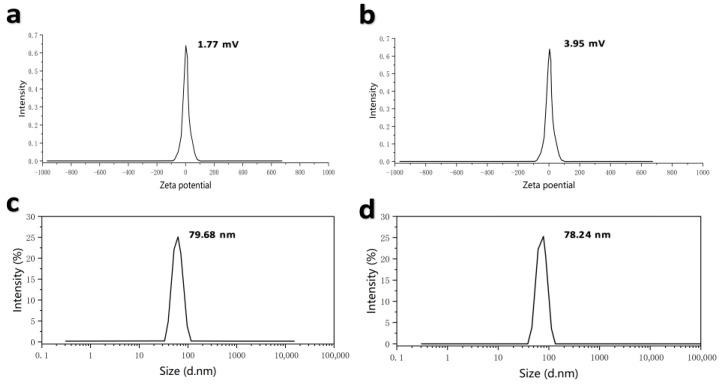
Detection of ZETA potential and particle size of XBS01 and XBS02 vaccine formulations. (**a**) ZETA potential detection results of XBS01. (**b**) ZETA potential detection results of XBS02. (**c**) Particle size detection results of XBS01. (**d**) Particle size detection results of XBS02.

**Figure 3 vaccines-13-01155-f003:**
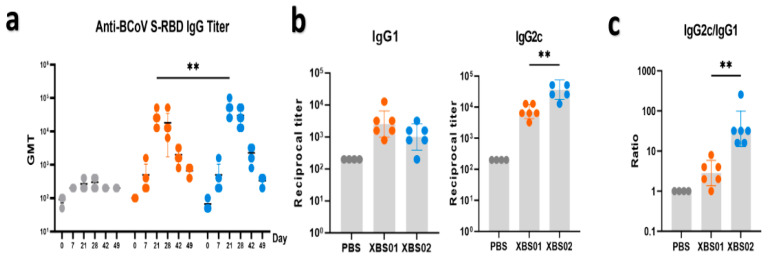
Anti-S-RBD antibody level detection. (**a**) Anti-S-RBD IgG titers were detected by ELISA, with endpoint titers shown. (**b**) On day 28, anti-S-RBD IgG1 and IgG2c titers were detected by ELISA, with endpoint titers shown. (**c**) The IgG2c/IgG1 ratio is presented. (n = 6, ** *p* < 0.01).

**Figure 4 vaccines-13-01155-f004:**
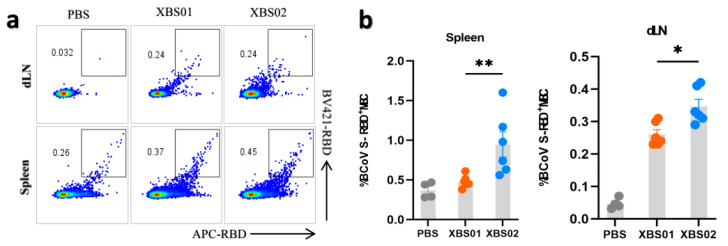
Memory B-cell responses induced by XBS01 and XBS02 in mice. BALB/c mice were given 10 μg mRNA vaccine by intramuscular injection on day 0 and day 14. Spleen and lymph node samples were collected 28 days after the second immunization. (**a**,**b**). Frequencies of class-switched S-RBD-specific memory B cells (MBCs) in draining lymph nodes and spleens were evaluated by flow cytometry. Data are presented as mean ± standard error of the mean (SEM). (n = 6, * *p* < 0.05, ** *p* < 0.01)

**Figure 5 vaccines-13-01155-f005:**
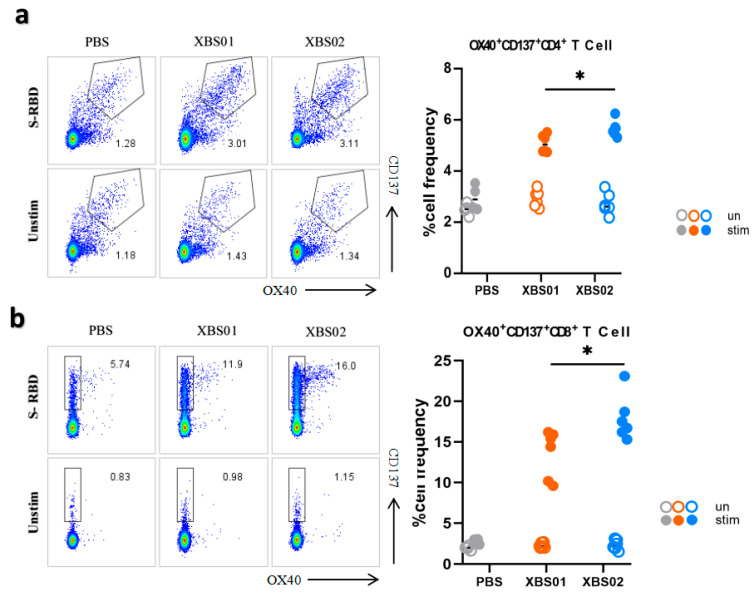
T-cell responses induced by XBS01 and XBS02 in mice. BALB/c mice were given 10 μg mRNA vaccine by intramuscular injection on day 0 and day 14. Spleen and lymph node samples were collected 28 days after the second immunization. (**a**) Splenocytes were stimulated with or without S-RBD antigen (5 μg/mL) for 20 h. The frequency of OX40^+^CD137^+^ T cells gated on CD3^+^/CD4^+^ was determined by flow cytometry. (**b**) Splenocytes were stimulated with or without S-RBD antigen (5 μg/mL) for 20 h. The frequency of OX40^+^CD137^+^ T cells gated on CD3^+^/CD8a^+^ was determined by flow cytometry. Data are presented as mean ± standard error (SEM). (n = 6, * *p* < 0.05).

**Figure 6 vaccines-13-01155-f006:**
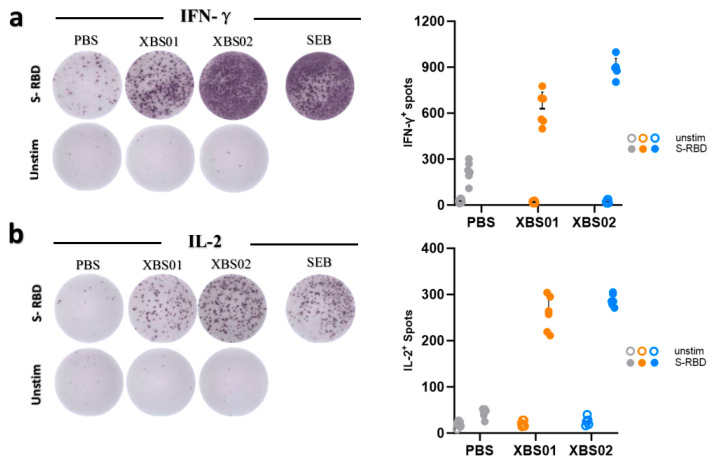
For the ELISpot assay, PBS served as the blank control, XBS01 and XBS02 as the experimental groups, and SEB as the positive control. (**a**,**b**) Immunocompetence and activation status of splenocytes were evaluated by the number of positive spots in ELISpot.

## Data Availability

Due to ethical restrictions, the raw data cannot be made publicly available. However, de-identified data may be obtained from the corresponding author upon reasonable request.

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
