# Peer review of "Preparation and Immunological Efficacy Evaluation of mRNA Vaccines Targeting the Spike Protein of Bovine Coronavirus"

_vaccines, 2025, doi:10.3390/vaccines13111155_

Round 1

Reviewer 1 Report

Comments and Suggestions for Authors

The objective of the present manuscript is to develop safe and effective preventive vaccines against BCoV, which is of immense significance. In this study, authors designed and prepared two mRNA vaccines that target the 28 Spike Protein Receptor-Binding Domain (S-RBD) of BCoV. The manuscript is well presented with clear data and clear figures. The title is clear and concise. The abstract is concise and focused. The objective is clear. Materials and methods are clear and written in detail.  The results are presented with clear and useful data and figures.  The discussion is clear and well presented. It would be beneficial if the author included information about the limitations and future steps. The references are sufficient and include recent sources from 2024. 

Author Response

Comments 1:It would be beneficial if the author included information about the limitations and future steps.

Response 1: Regarding the additional content you suggested, we have added it to the Discussion section (Lines 393-399). Thank you for your valuable comments on this manuscript.

Reviewer 2 Report

Comments and Suggestions for Authors

Abstract, reduce it, it is way too long

Introduction, add information regarding other platform vaccines against bovine coronavirus.

Line 131-133, "Two mRNA constructs targeting the BCoV S protein receptor-binding domain (S-RBD) were designed: XBS01 (linked to the Qα sequence) and XBS02 (linked to the IL-6 sequence). ", provide more information in this regard, all sequences must be provided, is the sequence capped? Poly-A tail?

Line 136-137, "synthesize mRNA, with T7 polymerase, N1-methylpseudouridine (m1Ψ), and Cap1 analogs", again, not enough info is provided for replication.

Line 137, "Purified RNA was resuspended in TE buffer to a volume of 50 μL", provide enough info for replication.

Line 138-139, "lipid components (cationic lipid:helper lipid:cholesterol:PEGylated lipid) were dissolved in ethanol at a molar ratio of 50:10:38.5:1.5", what cationic lipid? helper lipid? pegylated lipid? There a thousand options. Provide details.

Line 140-141, "lipid mixture was combined with mRNA dissolved in 10 mM citrate-sodium citrate buffer (pH 4.0) using a microfluidic device, with a nitrogen-to-phosphorus (N/P) ratio of 5.3:1", nitrogen from where? I guess from aminated lipids, but I should not be guessing.

Line 142, " After dilution with PBS", 2 times? 10 times?

Line 143, "was ultrafiltered using a 50 kDa ultrafiltration centrifugal filter", to remove ethanol? For washing? To disperse in PBS?

Line 143-145, "The particle size and encapsulation efficiency of the vaccine formulation were characterized using a NanoBrook Omni ZetaPlus", conditions? dilution? refractive indices? How was encapsulation efficiency measured using dynamic light scattering? Provide a detailed protocol for measuring encapsulation.

Line 182, "SEB", this has not been previously defined.

Figure 1, 2, and 3, run statistical analyses

Line 206, "with 10 μg/mL mRNA vaccines", according to 2.4, you injected mice with a vaccine having a concentration of 50 μg/mL.

Line 216, "with 10 μg/mL mRNA vaccines", same comment as previous.

Line 236-237, "Transmission electron microscopy", this was not reported in the methodology section, and the results are not reported.

Line 238-239, "56.01±1.6 nm (XBS01) and 70.26±1.6 nm (XBS02)", these error measurements are way to small. Is the reported error obtained from the preparation of three formulation replicates?

Discussion, compare your vaccines with other vaccines against the virus studied for different platforms (e.g. inactivated virus, viral vectored, etc). 

Author Response

Comments 1:
Abstract, reduce it, it is way too long
Response 1:
Thank you for your valuable comments on the manuscript. We have shortened the abstract (Lines 21-41) to less than 250 words.
Comments 2:
Introduction, add information regarding other platform vaccines against bovine coronavirus.
Response 2:
We have added content related to bovine coronavirus research (Lines 59-62). Since there are currently no commercialized vaccines for bovine coronavirus in China, we only reviewed relevant literature and listed two studies on bovine coronavirus vaccines.
Comments 3:
Line 131-133, "Two mRNA constructs targeting the BCoV S protein receptor-binding domain (S-RBD) were designed: XBS01 (linked to the Qα sequence) and XBS02 (linked to the IL-6 sequence). ", provide more information in this regard, all sequences must be provided, is the sequence capped? Poly-A tail?
Response 3:
We have only supplemented the disclosable Qα sequence and IL-2 sequence (Lines 134-136); the remaining sequences will not be disclosed temporarily. Please contact the corresponding author if necessary. A schematic diagram of mRNA design can be referred to GA, which has been added to the revised manuscript.
Comments 4:
Line 136-137, "synthesize mRNA, with T7 polymerase, N1-methylpseudouridine (m1Ψ), and Cap1 analogs", again, not enough info is provided for replication.
Response 4:
We have supplemented the in vitro transcription process (Lines 138-142) and added relevant details in the reagents section.
Comments 5:
Line 137, "Purified RNA was resuspended in TE buffer to a volume of 50 μL", provide enough info for replication.
Response 5:
We have provided the company of the RNA purification kit (Lines 142-143). Detailed experimental methods were omitted in the manuscript to reduce length; specific procedures can be performed according to the kit instructions, which is also how we conducted the experiment.
Comments 6:
Line 138-139, "lipid components (cationic lipid:helper lipid:cholesterol:PEGylated lipid) were dissolved in ethanol at a molar ratio of 50:10:38.5:1.5", what cationic lipid? helper lipid? pegylated lipid? There a thousand options. Provide details.
Response 6:
We have specified the types of lipids (Lines 145-146) and made corresponding revisions in the reagents section.
Comments 7:
Line 140-141, "lipid mixture was combined with mRNA dissolved in 10 mM citrate-sodium citrate buffer (pH 4.0) using a microfluidic device, with a nitrogen-to-phosphorus (N/P) ratio of 5.3:1", nitrogen from where? I guess from aminated lipids, but I should not be guessing
Response 7:
The nitrogen in the N/P ratio mentioned in the manuscript comes from cationic lipids. Since the chemical formula of the cationic lipid was not included in the manuscript, the N/P ratio has been deleted.
Comments 8:
Line 142, " After dilution with PBS", 2 times? 10 times?
Response 8:
We have supplemented the details as "diluted 10-fold with PBS" (Lines 148-149).
Comments 9:
Line 143, "was ultrafiltered using a 50 kDa ultrafiltration centrifugal filter", to remove ethanol? For washing? To disperse in PBS?
Response 9:
We have supplemented the purpose of ultrafiltration (Lines 149-151).
Comments 10:
Line 143-145, "The particle size and encapsulation efficiency of the vaccine formulation were characterized using a NanoBrook Omni ZetaPlus", conditions? dilution? refractive indices? How was encapsulation efficiency measured using dynamic light scattering? Provide a detailed protocol for measuring encapsulation
Response 10:
We have supplemented the experimental procedures for measuring encapsulation efficiency and the corresponding calculation method (Lines 151-165). For particle size measurement using dynamic light scattering, we have also added the operation details.
Comments 11:
Line 182, "SEB", this has not been previously defined.
Response 11:
We have replaced "SEB" with its full name "Staphylococcal Enterotoxin B" at its first occurrence (Line 203).
Comments 12:
Figure 1, 2, and 3, run statistical analyses
Response 12:
We have replaced the statistical graphs in the manuscript and added the sample size and P-values in the figure legends.
Comments 13:
Line 206, "with 10 μg/mL mRNA vaccines", according to 2.4, you injected mice with a vaccine having a concentration of 50 μg/mL.
Response 13:
The deviation in the mRNA dose was caused by a writing error. We have corrected the description of "10 μg/mL" in the figure legend.

Reviewer 3 Report

Comments and Suggestions for Authors

In this manuscript, the authors used multi-dimensional experimental methods (ELISA, memory B cell analysis, CD4⁺/CD8a⁺ T cell detection) and demonstrated the advantages of the two vaccines XBS01 and XBS02 in stimulating immune responses, with XBS02 being more effective. The manuscript is logically structured and written in good English. The experiments are beyond doubt. The manuscript may be accepted for publication after considering the reviewer's recommendations:

  1. It is necessary to write all affiliations with a capital letter and the country must be indicated.
  2. The reference format does not meet the journal's requirements. Some references, for example, 6 and 10, refer to the same article.
  3. Lines 74-77: «BCoV has five main structural proteins: nucleocapsid protein (N protein), membrane glycoprotein (M protein), envelope protein (E protein), hemagglutinin-esterase protein (HE protein), and spike protein (S protein) [9].»

There is no information on main structural proteins in ref. 9

  1. Lines 77-78: «The coronavirus S protein consists of three main regions: an extracellular domain, a transmembrane domain, and an intracellular domain [10].»

In ref 10 the E1 glycoprotein is described, not the S-protein.

  1. Lines 80-83: «BCoV invades host cells by binding the S1 subunit of its S protein to the host cell membrane receptor (N-acetyl-9-O-acetylneuraminic acid) [12], while the S2 subunit mediates the fusion of the viral envelope with the host cell membrane, enabling viral entry into the host cell [13].

There is no information in ref. 9 on BCoV invades host cells by binding the S1 subunit of its S protein to the host cell membrane receptor (N-acetyl-9-O-acetylneuraminic acid).

  1. Ref 13 should be «WANG Y H, GILBERT M, PERRAULT S. Coronaviruses: An Updated Overview of Their Replication and Pathogenesis [J]. Methods in Molecular Biology, 2020, 2203: 1-29»

The relevant information on page 6.

  1. Please, decipher the abbreviation SEB
  2. Figure 1 should be divided into two figures.
  3. Please, add what triangles in Figure 1a mean. For example, the reviewer did not find information on mice weight throughout the manuscript.
  4. Please, add figures and experimental details on transmission electron microscopy and dynamic light scattering analysis (lines 236-237).
  5. Explain in the manuscript how the encapsulation efficiency (line 239) was calculated.
  6. Figure 1a-d appears twice (page 6 and page 8). The same for figure1e-f on page 9
  7. Please, add data on PBS groups to the dot plots (Figure 2, on the left)

Author Response

Comments 1:It is necessary to write all affiliations with a capital letter and the country must be indicated. The reference format does not meet the journal's requirements.
Response 1:Thank you for your valuable revision suggestions on this manuscript. We have added the country to the authors' affiliations. Regarding the reference format, we are not familiar with handling link redirection, so we only supplemented the DOI numbers of the references in the manuscript.
Comments 2:Lines 74-77: «BCoV has five main structural proteins: nucleocapsid protein (N protein), membrane glycoprotein (M protein), envelope protein (E protein), hemagglutinin-esterase protein (HE protein), and spike protein (S protein) [9].» There is no information on main structural proteins in ref. 9;Lines 77-78: «The coronavirus S protein consists of three main regions: an extracellular domain, a transmembrane domain, and an intracellular domain [10].» In ref 10 the E1 glycoprotein is described, not the S-protein. Lines 80-83: «BCoV invades host cells by binding the S1 subunit of its S protein to the host cell membrane receptor (N-acetyl-9-O-acetylneuraminic acid) [12], while the S2 subunit mediates the fusion of the viral envelope with the host cell membrane, enabling viral entry into the host cell [13]. There is no information in ref. 9 on BCoV invades host cells by binding the S1 subunit of its S protein to the host cell membrane receptor (N-acetyl-9-O-acetylneuraminic acid).
Response 2:Due to my oversight, the citations in the manuscript did not match the references. We have now revised the reference section accordingly.
Comment3 3:Please, decipher the abbreviation SEB
Response 3:We have added the full name of SEB at its first occurrence in the manuscript.
Comments 4:Figure 1 should be divided into two figures. Please, add what triangles in Figure 1a mean. For example, the reviewer did not find information on mice weight throughout the manuscript.
Response 4:We have split the two original figures into four, each corresponding to the subsequent results. Thank you for pointing out the "body weight measurement" content in the immunization schedule part of the figure. We have supplemented the purpose of body weight measurement in the Discussion section (Lines 408-414).
Comments 5:Please, add figures and experimental details on transmission electron microscopy and dynamic light scattering analysis (lines 236-237).
Response 5:Possible errors in the school’s equipment or insufficient handling of details during our operation led to deviations in the sample particle size measurement. We have sent the samples to Seville Company for testing. Upon completion of the testing, we will immediately revise the data and add the corresponding figures and tables in the manuscript.
Comments 6:Explain in the manuscript how the encapsulation efficiency (line 239) was calculated.
Response 6:We have supplemented the calculation formula for encapsulation efficiency in the manuscript (Lines 161-162).
Comments 7:Figure 1a-d appears twice (page 6 and page 8). The same for figure1e-f on page 9. Please, add data on PBS groups to the dot plots (Figure 2, on the left)
Response 7:We have revised the figures and added the PBS group data to the dot plots as you requested.

Round 2

Reviewer 2 Report

Comments and Suggestions for Authors

From this previous comment (Line 131-133, "Two mRNA constructs targeting the BCoV S protein receptor-binding domain (S-RBD) were designed: XBS01 (linked to the Qα sequence) and XBS02 (linked to the IL-6 sequence). ", provide more information in this regard, all sequences must be provided, is the sequence capped? Poly-A tail?) onward, I do not see that the authors modified anything in the manuscript. 

Author Response

Comments 1:From this previous comment (Line 131-133, "Two mRNA constructs targeting the BCoV S protein receptor-binding domain (S-RBD) were designed: XBS01 (linked to the Qα sequence) and XBS02 (linked to the IL-6 sequence). ", provide more information in this regard, all sequences must be provided, is the sequence capped? Poly-A tail?) onward, I do not see that the authors modified anything in the manuscript.

Response 1: The full sequence has been submitted as supplementary material at the end of the manuscript, but it cannot be included in the main text. We apologize for any inconvenience. During sequence design, modifications including capping and poly-A tail addition were performed.

Reviewer 3 Report

Comments and Suggestions for Authors

Thank you for taking into account some of the reviewer's suggestions. However, for the manuscript to have scholarly value, the following reviewer's comments must be taken into account:

  1. The references are still not corrected (Refs. 9, 10, and12). The reviewer requests that fragments with relevant information from references 9, 10 and 12 will be added to "responces to referee".

2. The author`s response: "Possible errors in the school’s equipment or insufficient handling of details during our operation led to deviations in the sample particle size measurement. We have sent the samples to Seville Company for testing. Upon completion of the testing, we will immediately revise the data and add the corresponding figures and tables in the manuscript."

The reviewer doesn't understand why a revised version of the manuscript was submitted if the nanoparticle characteristics have not yet been clarified. The reviewer advises submitting a revised version after receiving the relevant data, including a description of the methods used and relevant figures/tables.

Typos:

Notably, mRNA vaccine technology has not yet been applied in ruminants  such as cattle, representing a promising development direction(,) and this research provides valuable theoretical guidance and practical references for future vaccine development, optimization, and clinical application. 

Author Response

Comments 1:The references are still not corrected (Refs. 9, 10, and12). The reviewer requests that fragments with relevant information from references 9, 10 and 12 will be added to "responces to referee".

Response 1:References 9, 10, and 12 have been revised as follows:

Reference 9: Modified content added to the manuscript: "ORF3, ORF4, ORF8, ORF9, and ORF10 encode hemagglutinin-esterase (HE) protein, spike (S) glycoprotein, envelope (E) protein, membrane (M) protein, and nucleocapsid (N) protein, respectively, which are components of virions." Reference 10: Modified content added to the manuscript: "The coronavirus spike contains three segments: a large ectodomain, a single-pass transmembrane anchor, and a short intracellular tail." Reference 12: Modified content added to the manuscript: "What is known is that the peripheral S1 portion can independently bind cellular receptors while the integral membrane S2 portion is required to mediate fusion of viral and cellular membranes." "The distribution of coronavirus receptors is critical to the pathogenic outcome. In this regard, it is notable that coronavirus spikes exhibit a range of receptor specificities;...... bovine coronaviruses recognize 9-O-acetylated sialic acids (Holmes and Dveksler, 1994)."

Comments 2:The reviewer doesn't understand why a revised version of the manuscript was submitted if the nanoparticle characteristics have not yet been clarified. The reviewer advises submitting a revised version after receiving the relevant data, including a description of the methods used and relevant figures/tables.

Response 2:We apologize for the inconvenience caused. Corresponding experiments were conducted during our research; however, the particle size data showed significant discrepancies due to outdated equipment. Since particle size is fundamental data for this vaccine, it was still included in the manuscript. Prior to submission, we prepared the same vaccine three additional times and sent the samples to a biotechnology company for testing. The results were only recently obtained. We have updated the results and figures in the revised manuscript, added the detailed experimental procedures, and highlighted the modified parts in red in the text.

Comments 3:Notably, mRNA vaccine technology has not yet been applied in ruminants  such as cattle, representing a promising development direction(,) and this research provides valuable theoretical guidance and practical references for future vaccine development, optimization, and clinical application.

Response 3:This content has been revised in the Discussion section of the manuscript. The entire paragraph has been refined, and the revised content is highlighted in red in the Discussion section.

Round 3

Reviewer 2 Report

Comments and Suggestions for Authors

All my comments were addressed by the authors. Despite this, there are a lot of typos and punctuation errors in their responses. For example:

Line 137-138, "sing a microfluidic device, The ratio"

Line 139-140:,"the formulation was ultrafiltered using a 50 kDa ultrafiltration centrifugal filter,To remove"

Line 155-157, "water, viscosity 0.8936 mPa・ s, refractive index 1.33; temperature: 25 C; electric field parameters: 40-50 V, 12 sub-tests"

Moreover, this information:

"Dilute the LNPs sample with PBS to a concentration of 1 mg/mL, pipette 1 mL of the diluted sample, slowly inject it into a test tube, then place the test tube into the instrument sample chamber, select the analysis parameters, and determine the particle size using a NanoBrook Omni ZetaPlus."

must be deleted since the authors already specified details of the analysis in the previous text (line 151-158).

Author Response

Dear Reviewer,

Thank you for your careful review and valuable comments, which have greatly helped improve the quality of our manuscript. We have carefully addressed all your suggestions as follows:

Regarding the typos and punctuation errors in our responses: We have thoroughly checked the entire text (including Lines 137-138, 139-140, 155-157) and corrected all identified errors.

Regarding the duplicate description of particle size analysis: We agree with your comment and have deleted the redundant content as requested.

We hope the revised manuscript meets the requirements. Thank you again for your time and efforts in reviewing our work.

Sincerely,

The Authors

Reviewer 3 Report

Comments and Suggestions for Authors

The authors took into account the reviewer's suggestions

Author Response

Dear Reviewer,

We hope the revised manuscript meets the requirements. Thank you again for your time and efforts in reviewing our work.

Sincerely,

The Authors